# Lipid and Lipoprotein Dysregulation in Sepsis: Clinical and Mechanistic Insights into Chronic Critical Illness

**DOI:** 10.3390/jcm10081693

**Published:** 2021-04-14

**Authors:** Grant Barker, Christiaan Leeuwenburgh, Todd Brusko, Lyle Moldawer, Srinivasa T. Reddy, Faheem W. Guirgis

**Affiliations:** 1Department of Emergency Medicine, College of Medicine-Jacksonville, University of Florida, 655 West 8th Street, Jacksonville, FL 32209, USA; grant.barker@jax.ufl.edu; 2Department of Aging and Geriatric Research, College of Medicine, University of Florida, Gainesville, FL 32603, USA; cleeuwen@ufl.edu; 3Department of Pathology, Immunology and Laboratory Medicine, College of Medicine, University of Florida Diabetes Institute, Gainesville, FL 32610, USA; tbrusko@ufl.edu; 4Department of Surgery, College of Medicine, University of Florida, Gainesville, FL 32608, USA; lyle.moldawer@surgery.ufl.edu; 5Division of Cardiology, Department of Medicine, David Geffen School of Medicine at UCLA, Los Angeles, CA 90095, USA; sreddy@mednet.ucla.edu

**Keywords:** sepsis, lipids, lipoproteins, chronic critical illness

## Abstract

In addition to their well-characterized roles in metabolism, lipids and lipoproteins have pleiotropic effects on the innate immune system. These undergo clinically relevant alterations during sepsis and acute inflammatory responses. High-density lipoprotein (HDL) plays an important role in regulating the immune response by clearing bacterial toxins, supporting corticosteroid release, decreasing platelet aggregation, inhibiting endothelial cell apoptosis, reducing the monocyte inflammatory response, and inhibiting expression of endothelial cell adhesion molecules. It undergoes quantitative as well as qualitative changes which can be measured using the HDL inflammatory index (HII). Pro-inflammatory, or dysfunctional HDL (dysHDL) lacks the ability to perform these functions, and we have also found it to independently predict adverse outcomes and organ failure in sepsis. Another important class of lipids known as specialized pro-resolving mediators (SPMs) positively affect the escalation and resolution of inflammation in a temporal fashion. These undergo phenotypic changes in sepsis and differ significantly between survivors and non-survivors. Certain subsets of sepsis survivors go on to have perilous post-hospitalization courses where this inflammation continues in a low grade fashion. This is associated with immunosuppression in a syndrome of persistent inflammation, immunosuppression, and catabolism syndrome (PICS). The continuous release of tissue damage-related patterns and viral reactivation secondary to immunosuppression feed this chronic cycle of inflammation. Animal data indicate that dysregulation of endogenous lipids and SPMs play important roles in this process. Lipids and their associated pathways have been the target of many clinical trials in recent years which have not shown mortality benefit. These results are limited by patient heterogeneity and poor animal models. Considerations of sepsis phenotypes and novel biomarkers in future trials are important factors to be considered in future research. Further characterization of lipid dysregulation and chronic inflammation during sepsis will aid mortality risk stratification, detection of sepsis, and inform individualized pharmacologic therapies.

## 1. Introduction

### 1.1. Sepsis Overview

Deriving from the ancient Greek word ‘sepo’ meaning “I rot”, the semantics of sepsis have proven nearly as complex as elucidating new treatments [1]. The current definition of sepsis (Sepsis-3) is a “life-threatening organ dysfunction caused by a dysregulated host response to infection,” represented by an increase in the patient’s Sequential Organ Failure Assessment (SOFA) score by 2 points or more [2]. The complexities of sepsis not only include dysregulated inflammation, but also alterations in endothelial microcirculation, metabolism, immune tissues, coagulation, microglial cells, and neurons [3].

According to the Centers for Disease Control and Prevention (CDC), at least 1.7 million adults in America develop sepsis each year, leading to 270,000 yearly deaths, which accounts for one in three in-hospital deaths [4]. Globally, there are an estimated 31.5 million sepsis, and 19.4 million severe sepsis cases, with 5.3 million deaths annually [5]. From a societal perspective, sepsis is one of the costliest in-hospital disease states to treat. In 2013, sepsis accounted for $24 billion in hospital expenses, representing 13% of US hospital costs, but only 3.6% of hospital stays [6]. Multiple studies have found the incidence of sepsis to be increasing, while the mortality rates have been decreasing [7,8,9]. The increase is commonly attributed to an aging population with more comorbidities, immunosuppressive use, drug-resistant pathogens, more frequent invasive procedures, use of implantable medical devices, changing definitions, and increased recognition [10,11]. These findings have recently been challenged due to changing diagnosis and coding practices, and the incidence and mortality rate may actually be relatively stable over the past decade [12]. Our group has focused on the ongoing morbidity and mortality following recovery from the acute septic phase.

### 1.2. Dysregulated Inflammation Driving Organ Failure, Chronic Critical Illness, and Death

As sepsis survival improves, disparate short and long-term outcomes have become more apparent. Analysis of Centers for Medicare and Medicaid Services (CMS) data from 2012–2018 found that those with sepsis-coded hospital admissions had three-times as many deaths as non-sepsis coded beneficiaries within one week of discharge, and more admissions to skilled nursing facilities [13]. These patients are more likely to die in these facilities, or face readmission to acute inpatient hospitals or transfer to nursing homes.

There are likely numerous contributors to poor long-term outcomes among sepsis patients. Persistent inflammation, immunosuppression, and catabolism syndrome (PICS) describes a subset of patients with chronic critical illness (CCI) who are characterized by persistent inflammation, reduced host immunity, ongoing organ injury, cachexia, metabolic derangements, and myeloid dysfunction [14]. CCI is defined as an intensive care unit (ICU) stay >14 days with ongoing organ dysfunction [15]. In PICS, elevations of the inflammatory cytokines including IL-6, IL-8, IL-10 characterize these patients. PICS is thought to initiate through two interrelated mechanisms. Firstly, immunosuppression results as a consequence of hematopoietic stem cell (HSC) expansion following the massive granulocyte demargination in early sepsis or trauma. This process is termed emergency granulopoiesis [16]. Mobilization of these stem cells leads to generation of a heterogenous group of immature myeloid cells known as myeloid derived suppressor cells (MDSCs) which exhibit T-cell suppressive activity [17]. Secondarily, viral reactivation, nosocomial infections, and MDSC infiltration of the kidneys and skeletal muscles drive the continuous release of pro-inflammatory mediators known as alarmins and damage-associated molecular patterns (DAMPs) [14,18].

While the exact mechanism driving MDSC proliferation in critical illness is unknown, experimental data in tumorigenesis and atherogenesis show a link with energy metabolism [19]. The ATP-binding cassette transporters A1 (ABCA1) and G1 (ABCG1) transport cholesterol from membranes to nascent and mature high-density lipoprotein (HDL), respectively, thereby regulating intracellular cholesterol homeostasis. Mice deficient in these transporters have a marked leukocytosis which is ameliorated through HDL-mediated prevention of HSC entry into the cell cycle, suggesting HDL can suppress myeloid cell proliferation independently of these receptors [20]. Synthetic HDL particles using human Apo-AI molecules have been shown to decrease MDSCs (in a murine model), and independently inhibit T-cell suppression by MDSCs [21]. Further research is needed to elucidate mechanisms and target interventions to prevent the development of CCI and PICS in sepsis patients, but lipids and their mediators clearly play important roles.

Despite comprising a minority of ICU patients, CCI patients account for a vastly disproportionate amount of ICU bed-days and costs, and have higher mortality [22]. At 12 months post-septic insult, these patients have a mortality eight times that of rapid recovery patients (40.1% vs. 4.8%) [23]. Not only is their long-term survival dismal, they also have significantly decreased physical function and health-related quality of life compared to rapid recovery patients [24].

## 2. Anti-Inflammatory and Protective Roles of Lipoproteins and Lipid Mediators in Sepsis

Patients undergo many distinct changes in their lipid profiles during states of acute physiologic stress. In addition to this, lipid mediators play a distinct role in innate immune signaling. The discovery and characterization of new molecules and processes provide new frontiers for targeting the septic response.

### 2.1. High-Density Lipoprotein

HDL is a complex biological molecule consisting mainly of two proteins known as Apo-A-I and Apo-A-II surrounding a lipid core. It is the smallest and densest of the lipoprotein molecules, and best characterized in reverse cholesterol transport, where excess cholesterol is transported from peripheral cells to the liver. HDL has a myriad of biological functions during inflammation described in Figure 1. Lipidomic studies have identified over 200 different lipid species in each non-pathologic HDL molecule [25].

In one study, HDL level at the time of emergency department (ED) admission in septic patients was more predictive of multiple organ dysfunction syndrome and 28-day mortality in receiver-operator curve (ROC) analysis than every other parameter measured, including lactate [28]. In another study of 64 patients with severe sepsis, no patient with an HDL >25 mg/dL on day 3 died, and the non-survivors had persistently declining HDL levels [29]. Total cholesterol levels follow a similar trend. HDL additionally undergoes qualitative changes, discussed later, which may contribute to further organ dysfunction and mortality.

### 2.2. Low Density Lipoprotein

Low-density lipoprotein (LDL) is molecule composed of a phospholipid monolayer containing a single apolipoprotein B-100 molecule associated with 80 to 100 additional proteins surrounding a hydrophobic core of polyunsaturated fatty acids (PUFAs) as well as esterified and unesterified cholesterol molecules. LDL is endocytosed by different tissues including the liver, through interaction of apo-B100 with the LDL receptor. Normal LDL lacks the multitude of immunoprotective functions as HDL, but plays an important role in endotoxin clearance. LDL, HDL, and very-low-density lipoprotein (VLDL) all bind and sequester lipopolysaccharide (LPS) from bacterial cell walls, although HDL has the highest affinity [30]. In addition to binding free LPS, LDL also facilitates release and binding of CD14-bound LPS from myeloid cells [31]. The LDL receptor (LDLR) later facilitates uptake of bound LPS as well as lipoteichoic acid (from gram positive bacteria) in a pathway negatively regulated by PCSK9 [32]. During inflammatory states, LDL becomes a dysfunctional and pathologic molecule termed oxidized LDL (oxLDL). Low LDL levels have been associated with higher long-term rates of community-acquired sepsis [33].

### 2.3. Specialized Pro-Resolving Mediators

Resolution of sepsis and inflammation is an active process mediated by specialized pro-resolving mediators (SPMs), a family of lipids discovered in the last decade that include lipoxins, resolvins, protectins, and maresins. Each of these compounds have their own novel biosynthesis pathways and receptors mediating physiological actions [34]. They directly mediate resolution of inflammation due to sepsis through effects on platelets, endothelium, and cyclooxygenase-2 (COX-2) expression [35]. The arachidonic acid derivatives known as prostaglandins and leukotrienes play an early role in vascular permeability and leukocyte chemotaxis [36]. As these mediators accumulate, prostaglandin E2 (PGE_2_) promotes a shift of eicosanoid biosynthesis towards lipoxin A_4_ (LXA_4_) generation, which opposes polymorphonuclear neutrophil (PMN) infiltration as well as functional responses [37]. Furthermore, LXA_4_ stimulates a concentration-dependent uptake and removal of apoptotic PMNs in tissue by monocyte-derived macrophages [38]. This pro-resolving process is dependent on PGE_2_, a product of cyclooxygenase-2 (COX-2), and inhibition of COX-2 activity has been shown to prolong inflammation rather than to promote resolution [39]. Sepsis non-survivors have elevated levels of certain SPMs, lipid mediators, and cytokines that suggest a failure of systemic resolution and possible failure or resistance of SPM receptors [35]. Appropriate phlogistic response is a delicate balance of both a suitable escalation of inflammation and its timely resolution; SPMs facilitate both. Our group is currently studying these mediators in a cohort of septic patients at 28–90 days post-sepsis to better characterize their role in the resolution of sepsis.

In addition to their promising anti-inflammatory role, SPMs have been found to enhance the native immune response. Resolvins in murine cecal ligation and puncture (CLP) models of sepsis reduce bacterial load, prevent excessive PMN activation, and enhance therapeutic effects of antibiotics [40,41]. Lipidometabolomic analysis of plasma from septic patients found sepsis non-survivors to have higher levels of proinflammatory prostaglandin F_2_ (PGF_2_) and leukotriene B_4_ (LTB_4_), as well as higher anti-inflammatory resolvins (RvE1, RvD1, RvD5, and Protectin D1 (PD1)) compared to survivors [35]. The mediators RvD2, RvE2, and LXB_4_ were associated with survival. These same species of resolvins show promise as important regulators in animal models of sepsis [40,41]. Increased levels of Protectin DX (PDX), an isomer of PD1 are highly predictive of ARDS (*p* < 0.001), even more so than APACHE II scores [35]. PD1 may also attenuate airway inflammation and hyperresponsiveness in human patients with asthma [42]. SPMs depend on a more complex network of ligand–receptor interactions than is currently understood. Lipid mediators exert their actions through G-protein coupled receptors (GPCRs) [43]. GPCR families are susceptible to downregulation, degradation, and signal modulation; all potentially affecting SPM interactions depending on the internal cellular environment.

## 3. Alterations of Lipid Metabolism in Sepsis Contribute to Failed Inflammation Resolution

It has long been known that humans and animals undergo significant changes to their endogenous lipid profiles during periods of critical illness [44]. Serum cholesterol levels drop by nearly 50% in severe sepsis [45]. HDL and LDL levels inversely correlate with levels of tumor necrosis factor *α* (TNF-*α*), IL-6, and IL-10 [29,46]. Multiple studies have shown hypocholesterolemia, especially of HDL, in septic patients on presentation and later in their clinical course to be predictive of non-survival [29,46,47,48]. Decreased hepatic synthesis and secretion of apolipoproteins caused by cytokines may contribute to critical illness-induced hypocholesterolemia, though causes and mechanisms have not been clearly delineated [49]. While inflammatory cytokines decrease hepatic lipoprotein synthesis, they also rapidly induce hepatic fatty acid (FA) and triglyceride synthesis, suppress fatty acid oxidation, and decrease triglyceride clearance by lipoprotein lipase [50,51,52,53,54,55]. Peripheral tissue lipolysis is stimulated during sepsis as well, further increasing plasma fatty acid levels [56]. Plasma fatty acid composition also undergoes significant alterations, with an overall decrease in polyunsaturated fatty acid (PUFA) levels and an increase in the ratio of omega-6 to omega-3 fatty acids [57]. Elevated plasma fatty acid levels in critical illness have been shown to decrease lymphocyte proliferation, promote neutrophil apoptosis and necrosis, and are associated with cardiac damage [58,59]. High triglyceride levels are an independent predictor of mortality in sepsis [60,61]. Intensive control of blood glucose has beneficial mortality effects in the critically ill (hypoglycemic events notwithstanding), with the greatest reduction of mortality in septic patients [62]. Normalization of the plasma lipid profile by insulin therapy through reduction in plasma FA level, and increases in plasma HDL and LDL are partially responsible for this benefit [63,64,65].

While the changes in lipid and lipoprotein metabolism during sepsis are well-characterized, the mechanisms underlying the lipid and lipoprotein changes are unknown. Genetic factors play a role. Gain-of-function variants in cholesterol ester transfer protein (CETP) in human studies have been associated with increased mortality from sepsis, and inhibition of CETP in humanized mouse models has been associated with preserved HDL levels and increased survival [66]. However, clinical trial data from cardiovascular trials have shown a concerning increase in severe infections among patients treated with the CETP inhibitor torcetrapib, an indicator of the metabolic complexity of lipid metabolism [67].

HDL levels are strongly determined by Apo-A-1 production and catabolism. Adiponectin levels, a protein hormone secreted by adipocytes, are inversely associated with Apo-A-1 catabolism, independent of obesity, insulin resistance, and HDL triglyceride content. Its levels are decreased significantly during sepsis as well as obesity [68,69]. While lower adiponectin levels are correlated with obesity, visceral fat loss has not been shown to significantly raise levels [70]. This is possibly related to differences in dietary and endogenous adipose fatty acid composition.

Low omega-6 to omega-3 ratios are associated with increased levels of adiponectin in obese and normal body mass index (BMI) patients [71,72,73] and the Western diet contains a high 15/1 ratio of omega-6 to omega-3 fatty acids [74]. Administration of omega-3 polyunsaturated fatty acids has been shown to raise serum adiponectin levels [75]. During the lipemia of sepsis, we propose the increase in omega-6 and omega-3 fatty acid ratio negatively affects the adipocytokine profile altering lipid and lipoprotein function [57]. Considering these factors and prior clinical trial data (see “Lipid Emulsions” section), we hypothesize administration of an omega-3 containing lipid emulsion will stabilize HDL levels in sepsis and favorably alter inflammatory profiles.

### Dysfunctional HDL

Though HDL is generally thought to be protective against sepsis, we and others have shown that when subjected to acute or chronic inflammation, HDL becomes prooxidant and proinflammatory [76,77,78,79,80]. In addition to quantitative changes during the acute phase response (APR), HDL undergoes reductions of enzymes critical for its function including lecithin cholesterol acetyltransferase (LCAT), cholesteryl ester transfer protein (CETP), paraoxonase (PON), and platelet-activating factor acetylhydrolase (PAF-AH) [81]. Replacing these molecules are proinflammatory serum amyloid A (SAA), Apo-J, and phospholipase A_2_ (PLA_2_) [82]. In addition to the alterations in the protein components of HDL, internal esterified cholesterol decreases and triglyceride content increases [83]. These changes seriously impair its immunomodulatory functions, converting it from an anti-inflammatory molecule to a proinflammatory one (Table 1).

The ability of HDL to inhibit oxLDL formation and LDL-induced monocyte-chemotactic activity (MCA) determines the HDL inflammatory index (HII) [108]. This is quantified using assays that fluoresce in the presence of LDL oxidation and MCA. In the absence of any HDL, the fluorescence level is normalized to 1.0. When the assays are incubated with test HDL, values >1.0 are termed dysfunctional HDL (dysHDL) and values <1.0 are anti-inflammatory. Our group has found dysHDL to be present in sepsis, correlate with adverse outcomes, and to predict the severity of organ failure [76,77]. Myeloperoxidase (MPO) is an enzyme most abundantly produced by neutrophils which catalyzes the generation of nitrating oxidants. These oxidants directly modify amino acid sites on ApoA-I, impairing its functional ability and creating dysHDL [79,80]. This oxidative damage limits the ability for lipid-free liberation of Apo-A-I from HDL, which is necessary to bind ABCA1, form nascent HDL, and clear cholesteryl esters from macrophages [78,109,110].

DysHDL fails to protect against accumulation of oxLDL, a type of LDL containing peroxidated lipids, lipoproteins, or their metabolites. OxLDL mediates leukocyte activation, proinflammatory cytokine secretion, leukocyte adhesion molecule expression, cellular degranulation, reactive oxygen species (ROS) release, and endothelial dysfunction contributing to cellular and tissue damage during sepsis [111]. OxLDL is bound by the lectin-like oxLDL receptor (LOX-1), which is found on endothelial cells, smooth muscle cells, intestinal cells, and macrophages. LOX-1 is upregulated by LPS, oxidative stress, and the presence of oxLDL [112]. LOX-1 deletion decreases mortality in murine sepsis [113]. This immune dysfunction caused by oxLDL is promoted through interactions with inflammasomes [114].

Inflammasomes are cellular protein complexes that comprise part of the innate immune system, and regulate the activation of caspase-1, to produce the mature form of the proinflammatory cytokine IL-1*β* [115]. The nod-like receptor pyrin domain containing 3 (NLRP3) inflammasome requires two signals for activation: priming by an NF-κB signal (such as LPS binding TLR4); and activation by ligands such as DAMPs and pathogen associated molecular patterns (PAMPs) [116]. It can also be activated by buildup of intracellular cholesterol [117]. NLRP3 plays a significant role in the respiratory, cardiac, renal, and central nervous system dysfunction found in sepsis [118]. Specific pro-resolving mediators have been shown to interact with and directly inhibit inflammasomes, thereby decreasing IL-1*β* production and the inflammatory response [119,120,121]. This reduces inflammatory cytokines that have been associated with increased mortality in sepsis [122].

The complex relationship between NLRP3 and SPMs is not yet fully understood. Ligand-receptor interactions may play a role in this complexity. For instance, the resolvin RvD2 is recognized by G-protein receptor 18 (GPR18), present on many cells including PMNs. Septic patients have less GPR18 expressed on PMNs and survivors have significantly higher expression than non-survivors [123]. The regulation of GPR18 expression is unknown. Synthetic RvD2 restores normal neutrophil migratory phenotype and improves survival after double-injury in burn/septic insult in mice [124].

SPMs are affected by lipoproteins and their associated enzymes. In vitro, healthy HDL attenuates macrophage proinflammatory lipid mediator production, and promotes pro-resolving mediator formation, notably RvD2 [125]. DysHDL not only fails to yield measurable RvD2 (among others) but also produces de novo proinflammatory LTB_4_. This provides a direct link between dysHDL and a dysregulated phlogistic response.

These pathways (Figure 2) play a role in acute sepsis, but also contribute to long-term organ dysfunction. Continuous low-grade inflammation such as diabetes, aging, and chronic illness lead to persistent low levels of circulating LPS termed low-grade endotoxemia. Even relatively low levels of endotoxin are sufficient to significantly impair HDL function [126]. Ongoing muscle catabolism and renal tubular injury causing DAMP release contribute to the pathogenesis and maintenance of CCI [14]. MDSC-related immunosuppression and latent viral reactivation continually feed this cycle of inflammation/impaired resolution as well.

## 4. Lipid-Based Therapies

Despite over 150 failed clinical trials, sepsis mortality has seen only modest improvement, mainly the consequence of earlier recognition and treatment, and reduced iatrogenic harm, rather than therapies directly modulating the septic response [3,127]. Subgroup analysis does occasionally demonstrate benefit, such as with afelimomab, granulocyte-macrophage colony-stimulating factor, anakinra, and trimodulin, but the clinical utility of these secondary data is unclear [128,129,130,131]. Over the past 20 years, lipid-based therapeutics and investigations into sepsis have greatly increased. The inflammatory and anti-inflammatory mechanisms of lipid mediators that have been elicited are mainly thanks to research into coronary artery disease and atherosclerosis. Lipids and innate immunity represent an enticing frontier in modulation of the septic response.

### 4.1. Statins

Statins are 3-hydroxy-3-methylglutraryl coenzyme A reductase inhibitors, whose primary therapeutic action is through decreasing mevalonate levels, a precursor for cholesterol synthesis, thereby decreasing LDL levels.

Interest in the therapeutic use of statins in critical illness began in the early 2000s due to their known anti-inflammatory effects through depletion of nonsterol cholesterol precursors regulating the inflammatory response. Following this reasoning, Liappis et al. first showed a significant reduction in mortality in bacteremic patients previously treated with statins [132]. It is important to note that the patients with prior statin use often are older and have a higher prevalence of comorbidities including obesity, diabetes, tobacco use, and ischemic heart disease [133]. Randomized controlled trials have failed to show the same mortality benefits as retrospective observational and cohort studies [134]. Wide study heterogeneity including characteristics of prior statin-users such as socioeconomic background, primary-care visits, and comorbidities confounds results, and correcting for confounding variables tends to diminish survival benefit [135]. Two single-center trials, Atorvastain in Reducing the Severity of Sepsis (ASEPSIS) and Hydroxymethylglutaryl-Coenzyme A Reductase Inhibition for Acute Lung Injury (HARP), both in statin-naïve patients, showed reductions in development of severe sepsis and SOFA score, respectively [136,137]. There remains opportunity to examine the role of statins in preventing critical illness progression in naïve patients.

Many lipid-independent mechanisms have been suggested for statins such as inhibition of T cell activation through inhibition of CD11a/CD18, antioxidant properties through the inhibition of superoxide production, inhibition of nitric oxide production which may prevent vasopressor-resistance [135], and even direct antimicrobial activity [138]. In the largest retrospective statin study performed to date, Lee et al. [139] examined the effect in sepsis of 52,737 patients previously treated with different statins. The authors used propensity score matching to match statin users with non-statin users and found both 30- and 90-day mortality benefit among users of atorvastatin and simvastatin, but not rosuvastatin. They conclude that the effects of statins on sepsis are not correlated to their lipid-lowering potency.

### 4.2. L-Carnitine

The primary biochemical function of carnitine is in the transport of long chain fatty acids through the carnitine acyltransferase system (CAT) (also called carnitine palmitoyltransferase) into the mitochondrial matrix for *β*-oxidation. Carnitine acyltransferase controls the acetyl CoA/CoA ratio, thereby controlling pyruvate metabolism and creating a reciprocal relationship between glucose and fatty acid oxidation [140]. This system is influenced during critical illness by LPS-induced alterations in CAT and a relative L-carnitine deficiency secondary to sepsis [141,142].

The Rapid Administration of Carnitine in SEPSIS (RACE) randomized clinical trial was a phase II clinical trial comparing 3 doses of L-carnitine vs placebo and its efficacy in reducing the Sequential Organ Failure Assessment (SOFA) score of patients with septic shock [143]. The primary outcomes were SOFA change from enrollment to 48 h, and 28-day mortality. The study was powered to predict a posterior probability of the high dose (18 g) levocarnitine being superior to placebo, and only reached 0.78 rather than the 0.9 a priori benchmark. These results are confounded by the multitude of sepsis phenotypes, and variability in patient drug response to L-carnitine supplementation, several of the authors have suggested pharmacologic probes using L-carnitine to target subgroups of septic patients more likely to respond to therapy [144]. The variability of these data highlight the need for further research into pharmacogenomics and personalized medicine approaches to sepsis therapeutics.

### 4.3. Lipid Emulsions

Enzymatic action on omega-6 fatty acids generates pro-inflammatory and anti-inflammatory lipid mediators through the arachidonic acid pathways. The omega-3 fatty acids eicosapentaenoic and docosahexaenoic acid (EPA/DHA), lead primarily to anti-inflammatory and cytoprotective pro-resolving lipid mediators discussed previously [145]. The primary biochemical evidence for the anti-inflammatory contribution of EPA is secondary to its derivative, the E-series resolvins [146]. These EPA/DHA derivatives function to regulate, enhance, and promote a self-limited immune response [147]. Omega-3 fatty acids have also been shown to inhibit the NF-*κ*B pathway and inhibit Toll-like receptor (TLR-4) activation [148]. Finally, EPA increases the functionality of HDL by increasing Apo-A1, thereby stabilizing PON, as well as directly activating HDL-associated PON, independently of Apo-A1 [149,150,151].

Modulation of pro-inflammatory and anti-inflammatory mediators by omega-fatty acid derivatives provides a clear therapeutic target in the treatment of sepsis. Meta-analyses of clinical trials have failed to demonstrate a consistent mortality benefit of omega-3 supplementation in sepsis, although several studies have shown a positive correlation between survival and omega-3 fatty acids [152]. Fish-oil in sepsis has led to reductions in ICU length of stay and days of mechanical ventilation as well [153]. A pilot randomized controlled trial (RCT) of a fish oil lipid emulsion in 60 ICU sepsis patients showed improved organ failure (first 7 days) and mortality (sub-population) compared to placebo, though results have not been replicated [154]. Inconsistent EPA/DHA concentrations, variable dosing schedules, as well as additional immunomodulating support may all confound data and repeatability [155].

The Lipid Infusion and Patient Outcomes in Sepsis (LIPOS) clinical trial used a phospholipid emulsion in gram-negative sepsis with the aim of clearing endotoxin and reducing 28-day mortality [156]. Results were negative, however, secondary analysis showed mortality benefit in patients with normal liver function and either total cholesterol >40 mg/dL or HDL >20 mg/dL [157]. Our group has previously shown in a phase I clinical trial, Lipid Intensive Drug therapy for Sepsis Pilot (LIPIDS-P) that omega-3 supplementation through a fish-oil containing lipid emulsion in early sepsis is safe, and promise toward stabilizing early total cholesterol and HDL levels [158]. A phase II clinical trial (LIPIDS-P) to further evaluate the efficacy is currently underway [98].

### 4.4. Eritoran

Eritoran is a synthetic lipid analogue that functions as a TLR-4 antagonist, downregulating LPS-induced stimulation of TNF-*α* and other inflammatory cytokines. Due to these effects on preventing overstimulation of the innate immune response, there has been clinical interest in its use in sepsis and influenza [159]. Preclinical studies demonstrated in vitro and in vivo benefit in halting LPS-induced alterations in cytokine levels or clinical signs and symptoms [160].

The ACCESS (A Controlled Comparison of Eritoran and placebo in patients with Severe Sepsis) trial was a phase 3 clinical trial of nearly 2000 patients with severe sepsis or septic shock treated with eritoran sodium vs. placebo. The primary end point was 28-day all-cause mortality, of which there was no significant difference between eritoran compared to placebo [161].

### 4.5. PCSK9

Reverse cholesterol transport is not only integral to physiological function, but also LPS clearance in gram negative sepsis. Several proteins including PLTP, scavenger receptors, and ABCA1 play dual roles in cholesterol and LPS clearance, suggesting an evolutionary role of lipid metabolism in ancestral innate immunity [162]. Proprotein convertase subtilisin/kexin type 9 (PCSK9) is an enzyme that promotes LDL receptor (LDLR) internalization and degradation, regulating its levels [163,164]. The LDLR removes cholesterol-rich LDL molecules from plasma. During states of infection, the hepatocytes clear LPS via the LDLR [165]. During states of acute inflammation, PCSK9 levels increase, increasing LDLR degradation [166]. Reduced LDL-clearance may have physiologic benefit as it has been shown that free cholesterol accumulation in macrophages independently activates TLR signaling [167]. Free cholesterol accumulation in macrophages has also been shown to increase TNF-*α* and IL-6 signaling independently of TLRs [168]. The indirect pro-inflammatory role of PCSK9 in infection make it an attractive target for modulation of the innate immune response in sepsis.

Secondary genetic analyses of larger sepsis trials have shown a correlation between lower plasma PCSK9, and PCSK9 loss of function with decreased sepsis mortality [169,170]. One observational cohort study found PCSK9 levels to positively correlate with organ failure [171]. No association has been shown between PCSK9 levels and risk of hospitalization for infection or sepsis [172]. To date, there are no human trials of PCSK9 inhibition in sepsis, and the only animal study of exogenous PCSK9 inhibition by antibodies to LPS challenge has not shown mortality benefit [173].

### 4.6. Fibrates

Peroxisome proliferator-activated receptors (PPARs) are ligand-activated transcription factors that control transcription of genes involved in metabolism, inflammation, proliferation, and differentiation. Endogenous PPAR ligands are mainly polyunsaturated fatty acids (PUFAs) and eicosanoids [174].

Decreased PPAR*α* levels have been shown to correlate with severity of septic shock in pediatric patients [175]. This is supported by increased bacterial load, decreased inflammatory cytokine levels (IL-6, TNF-a, IL-17 among others), and increased mortality in PPAR*α* null mice compared to wild type (WT) mice [175]. In rats, PPAR*α* agonist pretreatment has been shown to decrease TNF-a and other inflammatory cytokine levels [176]. PPAR*α* agonists have also been subsequently shown in vitro to decrease macrophage TNF-a, IL-6, NF-KB, and TLR4 expression [177].

Tissue-specific PPAR*α* expression is crucial for the metabolic shift from glucose to fatty-acid oxidation. In mice, hepatocyte PPAR*α* deletion decreases survival of *E. Coli* bacterial infection [178]. Animal studies have shown beneficial effects of PPAR*α* agonists in a cell-specific manner by ameliorating alveolar inflammation, preserving neutrophil migration ability, preventing myocardial dysfunction, and preventing endothelial damage [179,180,181,182]. To our knowledge, there are no human clinical trials of PPAR*α* agonists in sepsis. PPAR*α* activity has cell-specific, activation-specific, species-specific, and even gender-specific effects [183]. While it represents an attractive target, much research is needed before modulating PPAR*α* in human septic patients.

## 5. Discussion

In many instances, preclinical data have failed to translate meaningfully to human septic patients. Reasons for this vary, including heterogeneity of human sepsis phenotypes and limitations of animal models. In addition to the multitude of sepsis categories (that vary by infection type, source, and severity), retrospective analyses of critical illness databases have yielded new patient phenotypes of sepsis. In one of these, Seymour et al., elucidated four specific phenotypes (*α*, *β*, *γ*, *δ*) by analyzing a compilation of multiple cohorts yielding over 20,000 patients meeting Sepsis-3 criteria [184]. Each group has its own unique characteristics: *α*, most common, least vasopressor use; *β*, older, more chronic illness and renal dysfunction; *γ*, more inflammation and pulmonary dysfunction; and finally *δ*, which had more liver dysfunction and septic shock. Across all cohorts of trials, the *δ* phenotype had the highest 28-day and 365-day mortality.

Using these phenotypes, the authors ran simulation models of several RCTs. Simulation of the Protocol-based Care for Early Septic Shock (ProCESS) trial yielded the most notable results [185]. This clinical trial compared the central venous pressure and mean arterial pressure guidelines set forth in the original Early Goal Directed Therapy Trial (EGDT) with both a standardized resuscitation protocol and care at provider-discretion [186]. Simulation of the *δ* phenotype found EGDT to be harmful in over half, whereas simulation of the α phenotype actually found it to be superior. In the ACCESS trial, mentioned above, eritoran was found to be likely-harmful in δ phenotype patients.

Parallels can be drawn between the *δ* phenotype and others such as the inflammopathic cluster, Molecular Diagnosis and Risk Stratification of Sepsis (MARS) 2 cluster, and the sepsis response signature (SRS) 1 [187,188,189]. As transcriptomics and metabolomic characterizations evolve, these new biomarkers and metabolic signatures may yield more meaningful data when incorporated a priori in clinical trials.

The limitations of animal models of sepsis have been discussed extensively in other reviews and involve differences in type of septic insult, timing of insult, duration of therapy, supra-clinical doses of agents, pretreatment, exaggerated disease severity, variable transcriptomic responses, as well as differential metabolic and cardiovascular responses [190,191,192,193]. One of the most commonly-studied animals in preclinical research, the mouse, has a native lipid profile and metabolism that is markedly different from humans. Mice have lower overall cholesterol levels, predominantly carried in HDL molecules rather than LDL [194]. A natural deficiency of CETP in mice and reduced intestinal cholesterol uptake, partially explain this difference [195,196]. Transgenic human CETP mice exhibit dose-dependent decreases in HDL levels and an increase in VLDL and LDL levels, leading to a more ‘humanized’ lipid profile [197,198].

Most apparent is the murine resistance to atherosclerosis, a disease of low grade inflammation and dysregulated lipid metabolism. To model this in mice, they require Westernized diets and genetic modifications such as LDLR knockouts [199]. Using animal models with humanized lipid profiles may lead to more translatable clinical effects in studies of infection and inflammation in the future.

## 6. Conclusions

Lipid metabolism and innate immunity are closely linked and better understanding of their biology provides novel opportunities for targeted sepsis care. The heterogeneity between individual sepsis patients is a significant barrier to success in clinical trials. Temporal effects, secondary to patients presenting at different periods of illness, are a limiting factor as well. Signaling pathways require a ligand and a receptor, and variability in clinical response is not as straightforward as providing an exogenous ligand. These may have pleiotropic downstream effects when active at different points in cellular metabolism. Lipidomics is a powerful tool to provide temporal and phenotype-targeted therapies to improve sepsis survival and long-term outcomes. Our group seeks to use this information to better modulate host response in critical illness by improving early detection and prognostication, targeting patients likely to benefit, and lastly, preventing the immense burden of long-term sequelae secondary to PICS.

## Figures and Tables

**Figure 1 jcm-10-01693-f001:**
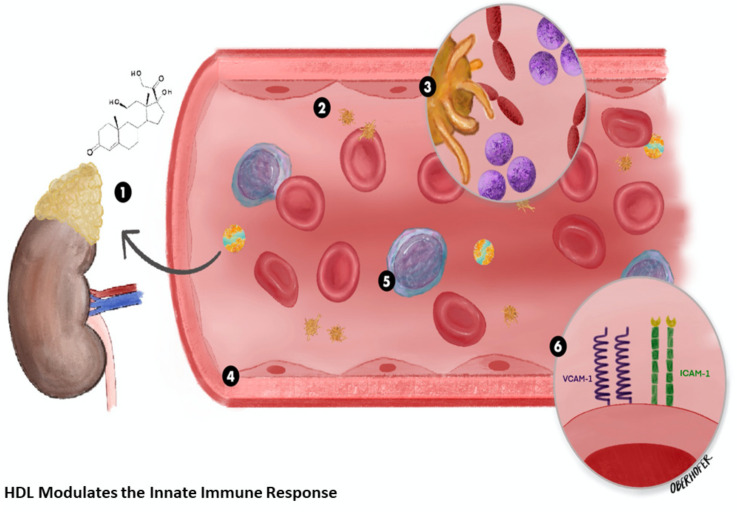
Functional high density lipoprotein (HDL) exhibits pleiotropic effects during an acute stress response: (**1**) Supports endogenous corticosteroid stress-response; (**2**) Decreases platelet aggregation; (**3**) Binds and clears bacterial toxins (lipopolysaccharide and lipoteichoic acid); (**4**) Inhibits endothelial cell apoptosis [26,27]; (**5**) Reduces the monocyte inflammatory response; (**6**) Inhibits expression of endothelial cell adhesion molecules vascular cellular adhesion molecule(VCAM)/intercellular adhesion molecule (ICAM)/E-selectin.

**Figure 2 jcm-10-01693-f002:**
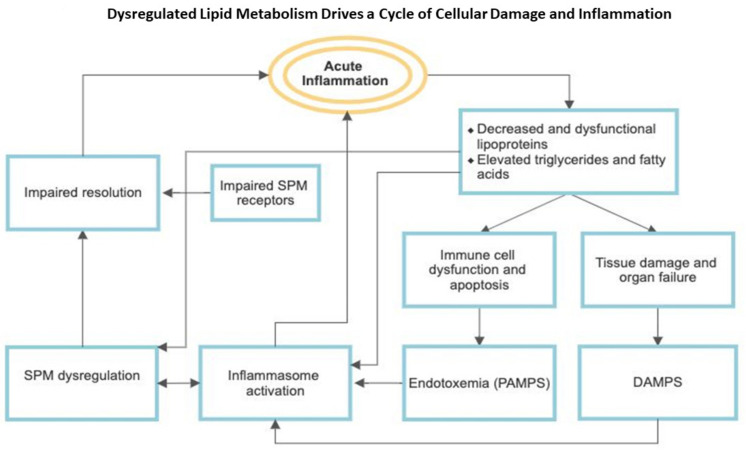
Dysfunctional HDL (Dys HDL) fails to regulate and actively impairs the innate immune response (see Figure 1). Elevations in triglycerides and fatty acids, which have innate signaling properties contribute to this impairment. This leads to tissue damage, organ failure, and immune cell dysfunction. These factors contribute to complex interactions between sustained inflammasome activation and dysregulated lipid pro-resolving mediators, maintained by low grade endotoxemia and DAMPs, leading to an inappropriate inflammatory response. SPMs, specialized pro-resolving mediators; PAMPs, pathogen-associated molecular patterns; DAMPs, damage-associated molecular patterns.

**Table 1 jcm-10-01693-t001:** Mechanisms of HDL dysfunction during sepsis by alterations in HDL-associated proteins.

Enzyme	Level	Function	Pathology during Sepsis
LCAT	↓	Promotes cholesterol efflux from cells to nascent HDL [84]	Diminished adrenal glucocorticoid function [85] Reduced LPS-neutralizing ability of HDL [86]
CETP	↓	Exchange of CE & TG between HDL and Apo-B-containing lipoproteins; promotes HDL maturation [87]	Missense variant in sepsis patients associated with HDL reduction, decreased survival, and increased organ failure [88]
PLTP	↑	Transfer of amphipathic molecules including phospholipids [89]	Regulates HDL size and composition [90] Recombinant PLTP in mice decreases bacterial growth and accelerates LPS detoxification [91]
PON	↓	Hydrolyzes lipid peroxides	Declines 71% in sepsis day 1–3 [92] Fails to inhibit oxLDL [93,94]
PAF-AH	↓	Hydrolyzes PAF	Declines 90% in sepsis day 1–3 [92] Failure to hydrolyze PAF, leading to immune cell activation, platelet activation, vascular permeability, and hypotension [95] Recombinant PAF-AH had no mortality benefit when used in septic patients [96]
EL	↑	Hydrolyzes HDL particles to liberate FFAs [97]	Upregulation leads to reduced HDL levels [98] Upregulation in inflammatory states may play a role in the resulting low-HDL state [99] EL knockout mice had increased survival to LPS-induced inflammation [100]
SAA	↑	Cytokine-like, propagates APR, modifies HDL transport [101]	>1000-fold increase during APR, displacing Apo-A-I [102] Comprises up to 80% of the proteins in higher-density HDL molecules [103] Increased HDL catabolism [104] Enhanced MDSC survival [105]
PLA_2_	↑	Hydrolyzes phospholipids to generate an FFA and lysophospholipid	Elevated lipoprotein-associated levels independent predictor of mortality in sepsis [106] Mainly mobilizes AA [107]

## Data Availability

Not applicable.

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
