# Peer review of "Lipid and Lipoprotein Dysregulation in Sepsis: Clinical and Mechanistic Insights into Chronic Critical Illness"

_jcm, 2021, doi:10.3390/jcm10081693_

Round 1
Reviewer 1 Report
This is a very interesting review of dysregulation of lipid and lipoprotein during sepsis. This is highly clinically significant area of research, especially, given the fact that as of today, there are no sepsis-specific therapies and sepsis continues to be the leading cause of death in the Intensive Care Units worldwide. The presentation of data/evidence is logical and thorough. There are some suggestions that will strengthen this manuscript further:
1). In places, the manuscript reads as a laundry list of lipids and lipoproteins that are dysregulated in sepsis. While this is a reflection of what is published in the literature, a better re-alignment of dysregulations leading to “pro-inflammatory” vs. “anti-inflammatory” phenotype may bring more clarity to the review.
2). While the manuscript strictly describes the pro- and anti-inflammatory phenotype perpetuated by the dysregulated lipid/lipoprotein metabolism, there is very little discussion regarding the “return of homeostasis”, which is an ultimate goal in sepsis and septic shock patients.
3). While it is clear from the review that the lipid and lipoprotein metabolism is dysregulated, there are no mechanistic insights regarding why this is happening. I understand that there is paucity of data in this field. However, this is a review article. Speculations/opinions are allowed and in fact thought provoking opinions are in fact welcome (in my opinion).
Author Response
Reviewer # 1
This is a very interesting review of dysregulation of lipid and lipoprotein during sepsis. This is highly clinically significant area of research, especially, given the fact that as of today, there are no sepsis-specific therapies and sepsis continues to be the leading cause of death in the Intensive Care Units worldwide. The presentation of data/evidence is logical and thorough. There are some suggestions that will strengthen this manuscript further:
1) In places, the manuscript reads as a laundry list of lipids and lipoproteins that are dysregulated in sepsis. While this is a reflection of what is published in the literature, a better re-alignment of dysregulations leading to “pro-inflammatory” vs. “anti-inflammatory” phenotype may bring more clarity to the review.
RESPONSE: We appreciate the reviewer’s comment and agree that in some places readability can be improved. Thus, we have edited the article for readability. Regarding the organization of the article, the current thematic structure is roughly as follows: 1) sepsis overview, 2) the role of inflammation in promoting and propagating organ failure, CCI, and death, 3) the role of lipids in the innate immune response (ie, normal lipid function), 4) altered lipid metabolism that leads to dysregulation and failed inflammation resolution (failed normal function of lipids), 5) lipid therapies, 6) discussion. Given the complexity and interrelatedness of pro and anti-inflammatory mediators, I think it would be difficult to break up the article in this manner. However, to add clarity, we have adjusted the section subtitles to give the reader a better sense of article flow.
2). While the manuscript strictly describes the pro- and anti-inflammatory phenotype perpetuated by the dysregulated lipid/lipoprotein metabolism, there is very little discussion regarding the “return of homeostasis”, which is an ultimate goal in sepsis and septic shock patients.
RESPONSE: Your insight is appreciated. We agree that this should be further highlighted in this paper. There is a discussion of the normal course of inflammation resolution in the pro-resolving mediator section. We have modified this section for increased clarity and to better reflect this. We also discuss the role of continual inflammation inhibiting the homeostatic return and our hypothesis for its non-resolution. We are currently researching lipid and lipoprotein profiles post-sepsis at 28 and 90 days in order to further elaborate this topic and have added this to the paper as well.
3). While it is clear from the review that the lipid and lipoprotein metabolism is dysregulated, there are no mechanistic insights regarding why this is happening. I understand that there is paucity of data in this field. However, this is a review article. Speculations/opinions are allowed and in fact thought provoking opinions are in fact welcome (in my opinion).
RESPONSE: Thank you for the suggestion. We are interested this area as well, and are happy to discuss this further. We have added to the sections “Alterations of Lipid Metabolism in Sepsis” our opinions regarding genetic and epigenetic factors, as well as adipokines and the profile of stored adipose tissue. Additionally, the mechanism of dysregulation of HDL and LDL in terms of oxidative stress are partially discussed in the section “Dysfunctional HDL” and “LDL”.
Reviewer 2 Report
The work presents an innovative feature and that received little discussion treated in the bibliography on sepsis. However, the contribution of this topic is scarce. The topics to be discussed are exhaustive and correct. Nevertheless, if we consider it as a narrative review, it can be interesting, although I miss considering the main characteristics of the articles, commenting on their strengths and weaknesses, as well as a classification of the quality of the selected articles, the reason and the cause of exclusion of others. If we consider it as a review -metanalysis- all the corresponding statistical treatment would be missing in addition to the aforementioned.
This article might be appropriate as a book chapter but not as a systematic review or meta-analysis.
Author Response
Reviewer # 2
The work presents an innovative feature and that received little discussion treated in the bibliography on sepsis. However, the contribution of this topic is scarce. The topics to be discussed are exhaustive and correct. Nevertheless, if we consider it as a narrative review, it can be interesting, although I miss considering the main characteristics of the articles, commenting on their strengths and weaknesses, as well as a classification of the quality of the selected articles, the reason and the cause of exclusion of others. If we consider it as a review -metanalysis- all the corresponding statistical treatment would be missing in addition to the aforementioned.
This article might be appropriate as a book chapter but not as a systematic review or meta-analysis.
RESPONSE: Thank you for your feedback. We wrote this paper as a literature review with current updates and progress regarding lipids in sepsis. It is our goal to present mechanistic insights into the role of lipids in sepsis and the effects of their dysregulation with a particular focus on lipid-based therapies as well as the role of lipids in the development of the persistent inflammation, immunosuppression, and catabolism syndrome (PICS). As this is paper does not involve a specific question, a systematic review would limit the breadth of data to present. Many of the listed therapies are currently theoretical and without human trials (PCSK9, Fibrates), we discuss the opportunity and reasoning for further study. This section has discussion and critique of multiple human trials as well. Additionally, the discussion section contains an analysis of the limitations of current human and animal trials and why preclinical research has failed to translate meaningfully.
Round 2
Reviewer 1 Report
The authors have addressed all the issues
Reviewer 2 Report
Changes should be appropriate.